# The Surplus Transplant Lung Allocation System in Italy: An Evaluation of the Allocation Process via Stochastic Modeling

**DOI:** 10.3390/ijerph18137132

**Published:** 2021-07-03

**Authors:** Corrado Lanera, Honoria Ocagli, Marco Schiavon, Andrea Dell’Amore, Daniele Bottigliengo, Patrizia Bartolotta, Aslihan Senturk Acar, Giulia Lorenzoni, Paola Berchialla, Ileana Baldi, Federico Rea, Dario Gregori

**Affiliations:** 1Unit of Biostatistics, Epidemiology and Public Health, Department of Cardiac, Thoracic, Vascular Sciences and Public Health, University of Padova, Via Loredan 18, 35121 Padova, Italy; corrado.lanera@unipd.it (C.L.); honoria.ocagli@unipd.it (H.O.); daniele.bottigliengo@studenti.unipd.it (D.B.); patrizia.bartolotta@studenti.unipd.it (P.B.); giulia.lorenzoni@unipd.it (G.L.); ileana.baldi@unipd.it (I.B.); 2Thoracic Surgery Division, Department of Cardiac, Thoracic, Vascular Sciences and Public Health, Padova University Hospital, Via Giustiniani 2, 35128 Padova, Italy; marco.schiavon@unipd.it (M.S.); andrea.dellamore@aopd.veneto.it (A.D.); federico.rea@unipd.it (F.R.); 3Department of Actuarial Sciences Beytepe, Ankara 06800, Turkey; aslihans@hacettepe.edu.tr; 4Department of Clinical and Biological Sciences, University of Torino, Regione Gonzole 10, 10043 Orbassano, Italy; paola.berchialla@unito.it

**Keywords:** probability model, surplus lungs, shiny app, protocol, excess organs

## Abstract

Background: Lung transplantation is a specialized procedure used to treat chronic end-stage respiratory diseases. Due to the scarcity of lung donors, constructing fair and equitable lung transplant allocation methods is an issue that has been addressed with different strategies worldwide. This work aims to describe how Italy’s “national protocol for the management of surplus organs in all transplant programs” functions through an online app to allocate lung transplants. We have developed two probability models to describe the allocation process among the various transplant centers. An online app was then created. The first model considers conditional probabilities based on a protocol flowchart to compute the probability for each area and transplant center to receive each n-th organ in the period considered. The second probability model is based on the generalization of the binomial distribution to correlated binary variables, which is based on Bahadur’s representation, to compute the cumulative probability for each transplant center to receive at least nth organs. Our results show that the impact of the allocation of a surplus organ depends mostly on the region where the organ was donated. The discrepancies shown by our model may be explained by a discrepancy between the northern and southern regions in relation to the number of organs donated.

## 1. Introduction

Since the first successful lung transplants were performed in the 1968, lung transplantation has become an established surgical treatment for chronic end-stage respiratory diseases [1]. In the United States the number of lungs transplanted has reached its highest level in 2019, with reduced waitlist mortality as reported by the Organ Procurement Transplantation Network (OPTN)[2]. Furthermore, in Italy the trend of lung transplants has increased until 2019 [3].

Given the scarcity of lung donors, their allocation is an issue that has been addressed with different strategies worldwide. Optimizing the allocation of donated organs is crucial to reduce the number of patients dying on the waiting list for lung transplantations (LTx) and to improve post-transplant outcomes [4]. Organ allocations rules are different across the world, as reported by the International Society for Heart and Lung Transplantation (ISHLT) [5]. Several countries have introduced the use of a scoring system to maximize the clinical matches between the tissue type of the donor and that of the recipient. The most used worldwide is the lung allocation score (LAS) [6], which was introduced in 2005 in the United States by the OPTN [7]; it gives a score to each candidate based on medical urgency and expected post-transplant survival [8]. In the US, the lung allocation system is now changing: In the continuous distribution model, they are incorporating the LAS and the geographic distribution [7]. A modified version of the LAS was introduced in 2009 in Europe by Eurotransplant, a network that consists of Austria, Belgium, Croatia, Germany, Hungary, Luxembourg, Netherlands, and Slovenia [9]. The utilization of this scoring system has achieved an initial decline in waiting list times and an increase in lung transplantation rates [7,10]. Other European countries have preferred to introduce different priority algorithms to reduce waiting list times and mortality rates, especially in patients with critical conditions. In 2007, a procedure of high emergency lung transplantation (HELT) was implemented in France [11]. Similar procedures, with different inclusion and exclusion criteria, was introduced in Spain, UK, and Nordic European countries [9]. These countries collaborate through the Scandiatransplant urgent lung allocation system (ScULAS), which is a system introduced in 2009 giving supranational priority to patients considered urgent [12].

In Italy, over 2660 lung transplants were reported from 1993 to 2020 by the Transplant Information System (SIT) [13]. The European LAS was introduced on a provisional basis on 15 March 2016 in Lombardia, a northern Italian region. Three lung transplantation centers are located in this region; the centers’ waiting lists were merged into a single regional waiting list, through which organs were allocated according to the LAS score [14,15]. The introduction of the LAS score has decreased the median waiting time and mortality in the list and increased the transplantation rate from 25% to 38% (*p* = 0.001) [14]. Other Italian centers are based on a “center decision” to allocate lungs on a clinical basis. Each center has a list of patients that require lung transplantation and a patient could be enlisted only in one center. Each center follows a rotation scheme to allocate lungs to its transplantation center [15]. Since 2010, Italy was introduced the Urgent Lung Transplant program (ULTp) managed by the Centro Nazionale Trapianti (CNT). ULTp serves to identify recipients “requiring lung transplant priority” at a national level [16,17]. When no urgent transplant cases are reported on a national level and the available organs are not used within the region where the donor is located, the organs are considered “surplus” and are allocated at a regional level according to a program developed by the CNT in 2014 [18]. For ethical reasons and the long-term sustainability of the system, the Italian allocation system tries to equally allocate organs among transplant centers.

This work aims to describe the “national protocol for the management of excess organs in all transplant programs” [18] through mathematical models. We also provide an interactive visualization tool for evaluating whether the current Italian policy on organ distribution can potentially result in distortions or inequalities in organ allocation systems across different areas of the country.

## 2. Materials and Methods

### 2.1. The Italian Surplus Transplant Protocol

The Italian system is a hybrid. Initially, the system allocates the organ according to priorities in the urgency list and then according to the geographical location of the donor. The latter criterion affects health-care resources and policies by increasing the workload for the CNT at all levels [18]. The Italian transplant system is coordinated by the CNT, which directly responds to the Ministry of Health. The CNT coordinates both the regional and interregional centers. When the organ is not allocated through the national program of urgency and it is not used in the location of the donor, the organ is considered “surplus” and is allocated according to the “national protocol for the management of excess organs in all transplant programs” [18]. This system allocates the surplus organ based on dividing the national territory into two “macroareas” (MAs), which are MA Center-North and MA Center-South. MA Center-North includes the following: regional transplant center (CRT) Sardegna, CRT Piemonte, CRT Emilia Romagna, CRT Toscana, and the CRT North Italian Transplant program. MA Center-South includes the following: CRT Lazio, CRT Calabria, CRT Basilicata, CRT Abruzzo-Molise, CRT Umbria, CRT Campania CRT Puglia, and CRT Sicilia (Figure 1). There is one CRT in each region, with the exception of the North Italian Transplant program (NITp), which includes several CRTs.

Therefore, when there are no urgent cases and the CRT of the donor has no organ requests, the organ is, firstly, at the disposal of the donor’s MA according to a “strip continuous model” (Table 1).

In this model, the CRTs of the MA are listed in a specific order in the respective strip. The order in the strip is updated according to the choice made in the previous allocation of a surplus organ. When a center accepts a surplus organ, in the subsequent distribution of a surplus organ, the center will be sent to the end of the list and the same rule applies for those CRTs that have rejected the organ previously. This system helps to distribute the organs equally and theoretically does not consider the importance of the center in terms of the number of transplants needed per year. In the allocation process, each CRT of the donor’s MA will receive the offer of the organ simultaneously, with an indication of its position in the strip. The allocation process is shown in detail in Figure 1.

For each donor’s CRT, the pathway of the organ is specified according to the protocol indications. If the organ is rejected by all the CRTs of the donor’s MA, it will be offered to the other Mas [18,19]. An example of how the system works is illustrated in the following scenario: a surplus organ is found in the CRT of the Toscana region, which belongs to the MA Center-North. The organs are at first offered to the CRTs of the MA-North (excluding Toscana since it is the one that has produced the exceeding organ and therefore does not need it) according to the continuous strip model. In the order shown in Table 1, the CRTs to which the offer is made are those of Piemonte, Emilia Romagna, and NITp. If none of these transplant centers require organs, it will be offered to the MA Center-South according to the order in the relative strip. In our example, the organ could be accepted at first by the CRT of the region Lazio and then, only if they refuse, it is offered to the CRT of the region Sicilia. If the CRT of Lazio does not accept the organ, it is definitively lost. The same mechanism occurs if the surplus organ is produced by a region that does not have a CRT for lung transplants, such as Valle d’Aosta (Figure 1).

### 2.2. Probability Models

We developed two probability models to describe the allocation process for lung transplants; the first model considers the probability of a center being offered and accepting the n-th lung, while the second model considers the probability of a center being offered and accepting at least n-th lungs.

By the established protocol, a region that produces an excess organ cannot also be required to use it. Hence, it is possible to compute the probability that a surplus organ is actually used by adding the probability for each MA to be selected and then to ask its CRTs to use the organ and the probability of it being accepted by at least one of them (excluding the one that produced the organ). The first model considers conditional probabilities based on the protocol flowchart (Figure 1) and computes the probability for each CRT (as defined in the protocol) of receiving the n-th organ in the period considered. The model, therefore, takes into account the following information: regions with or without CRTs, the configuration of the MA, and the probabilities (a priori, e.g., historical) for each CRT of accepting and using an excess organ offered and the current state of the continuous strip. The model for each option stores the information on the corresponding update status on the continuous strip to use it for the computation of the probabilities for the next (n+1)-th surplus organ.

The second probability model is based on the generalization of the binomial distribution with correlated binary variables based on Bahadur’s representation [20]. The model considers all the probabilities of receiving at least *k* organs out of the *N* provided during the period considered and based on the probability computation of the first model.

The probability models are described extensively in the Appendix A.

#### 2.2.1. Variable’s Convention

**Definition** **1.***Let* A*be the set of macroareas in the state. For our purpose and for all the probability models actually considered, that set contains just two elements, e.g., “North” and “South”.**For* δ ϵ A*, let* $n_{\delta }$*be the number of regions with at least one transplant center in macroarea* δ *and* $m_{\delta }$ *be the number of regions without any transplant centers in macroarea* δ.*Let* $C^{\delta }=\left\{c_{i_{\delta }}^{\delta }:i_{\delta }\in \left\{1,\ldots ,n_{\delta }\right\}\right\}$*be the set of (macro)regions with at least one transplant center in macroarea* δ.*Let* $O^{\delta }=\left\{o_{i_{\delta }}^{\delta }:i_{\delta }\in \left\{1,\ldots ,m_{\delta }\right\}\right\}$*be the set of regions without any transplant center in macroarea* δ.*Let* $P^{X}$*be the probability that a surplus organ is provided by a region* x∈X*where* X*is simply a generic set of regions. Define similarly* $P^{x}$*as the probability that a surplus organ is provided by a (supposed and well defined) region x.**Let* $P_{X}$*be the probability that if a surplus organ is offered to the set of region X, it will be used by some* x∈X*(without any consideration of the criteria leading the decision of which* x∈X*uses it). Define similarly* $P_{x}\;$*as the probability that a surplus organ offered to a (supposed and well defined) region x is accepted by x itself.*

**Proposition** **1.** **(overall** **probability** **of** **using** **a** **surplus** **organ).**
*With the notation in Definition 1, the probability for a surplus organ to be used is described as follows.*

$$\underset{\delta \in A}{\sum }[P_{C^{\delta }}\left[P^{O^{\delta }}+P^{O^{1-\delta }}\left(1-P_{C^{1-\delta }}\right)+\underset{i\in n_{\delta }}{\sum }P^{c_{i_{1-\delta }\;}^{1-\delta }}\left(1-P_{C^{1-\delta }\backslash c_{i_{1-\delta }\;}^{1-\delta }}\right)\right]+\underset{i\in n_{\delta }}{\sum }\left(1-P_{C^{\delta }\backslash c_{i_{\delta }\;}^{\delta }}\right)]\;$$


#### 2.2.2. Macroarea Level

**Definition** **2.***Let*
 M
*be the number of surplus organs supposed to exist in the period.**Let*  t∈{1,…,M}*be the t-th surplus organ provided in the period.**Let* $\;T_{c_{i_{\delta }}^{\delta }}^{t,j_{\delta }}$*be the probability for the (macro)region* $c_{i_{\delta }}^{\delta }$*to be at the position* $j_{\delta }\in \left\{1,\ldots ,n_{\delta }\right\}$*of its strip while the organ t-th (out of M) is provided.**Let* $\;P_{c_{i_{\delta },j_{\delta }}^{\delta }}$*be the probability for the center* $c_{i_{\delta }}^{\delta }$*to obtain (accepting it) a surplus organ if it is located at position*  $j_{\delta }$*in its strip.**Let*  $S_{x}$*be the number of surplus organs provided by a region x.*

**Proposition** **2.****(Probability for at least k surplus organs).***For every* n∈N*, let* $\;x^{n}=\{\begin{array}{c} \left(x_{1,\ldots },x_{n}\right):x_{i}\epsilon \left\{0,1\right\} \end{array}$*for every* i ∈{1,…,n}*be the set of all possible sequences of n 0s and 1s. Let*  $\zeta \left(x^{n}\right)=\overset{n}{\underset{t=1}{\sum }}x_{t}$*. Hence, for every (macro)region*
 $c_{i_{\delta }}^{\delta }$
*in a macroarea*  δ ϵ A 
*and for every number*
*0 < k ≤ M, the probability for* $c_{i_{\delta }}^{\delta }$
*to obtain at least k surplus organs out of M in the period considered is described as follows.*
$${}_{k}^{M}P{}_{c_{i_{\delta }}^{\delta }}=\underset{x:\zeta \left(x^{M}\right)>k}{\sum }\left[\overset{M}{\underset{t=1}{\prod }}\left[{\left(P_{c_{i_{\delta }}^{\delta }}^{t}\right)}^{x_{t}}{\left(1-P_{c_{i_{\delta }}^{\delta }}^{t}\right)}^{1-x_{t}}\right]\right]$$


#### 2.2.3. Macroregion Level

**Definition** **3.****(Variable’s convention for macroregion).** *Let R be the set of macroregions in the nation.**For every*  τ∈R*, let* $n_{\tau }$*be the number of regions with at least one transplant center in microregion**τ and* $m_{\tau }$*be the number of regions without any transplant centers in microregion τ. Moreover, let* $\bar{n_{\tau }}$*be the length of the strip of τ.**Let*  $C^{\tau }=\left\{c_{i_{\tau }}^{\tau }:i_{\tau }\in \left\{1,\ldots ,n_{\tau }\right\}\right\}$*be the set of regions with at least one transplant center in microregion τ.**Let*  $I_{c_{i_{\tau }}^{\tau }}\subset \left(1,\ldots ,,\bar{n_{\tau }}\right)$*be the starting set of indexes in which* $c_{i_{\tau }}^{\tau }$*appears in the strip of τ.**Let*  $I_{c_{i_{\tau }}^{\tau }}^{t}$*the set of indexes in which* $c_{i_{\tau }}^{\tau }$*appears when the t-th surplus organ is provided.**Let* $O^{\tau }=\left\{o_{i_{\tau }}^{\tau }:i_{\tau }\in \left\{1,\ldots ,m_{\tau }\right\}\right\}$*be the set of regions without any transplant centers in the macroregion τ.**For every*  τ∈R *let*  $S_{\tau }=\underset{C_{\tau }^{\;}}{\sum }S_{C_{\tau }}+\underset{O_{\tau }^{\;}}{\sum }S_{O_{\tau }}$.

**Proposition** **3.****(Probability of obtaining an organ at a given strip position in a macroregion)**. *For every*  τ∈R*and every* $i_{\tau }\in \left\{1\ldots ,n_{\tau }\right\},\;j_{\tau }\in \left\{1\ldots ,\bar{n_{\tau }}\right\}$*, the probability for the center* $c_{i_{\tau }}^{\tau }$*to obtain (accepting it) the t-th surplus organ if it is located at the position* $j_{\tau }$*in its strip is provided by its probability of accepting a surplus organ if it is asked, by multiplication of the probability that the organ is not produced in the region, and by the product of the probability that every region possibly before it in the strip does not accept the organ. It is described as follows.*$$P_{c_{i_{\tau },j_{\tau }}^{\tau }}^{t}=P_{c_{i_{\tau }}^{\tau }}\bar{S_{c_{i_{\tau }}^{\tau }}}\cdot \underset{c^{\tau }:I_{c^{\tau }}^{t}\cap^{\;}\left\{0,\ldots ,j_{\tau }-1\right\}\neq 0}{\prod }\left(1-P_{c^{\tau }}\right)$$

**Proposition** **4.****(Probability of obtaining a given surplus organ in a macroregion).** *For every surplus organ**t* ∈ {1,…, M}*provided, for every region* $c_{i\tau }^{\tau }\in \;C^{\tau }$*in a macroregion* τ ∈R*, the probability for* $\;c_{i_{\tau }}^{\tau }$*to obtain and accept the t-th surplus organ is described as follows*.
$$P_{c_{i_{\tau }}^{\tau }}^{t}=\overset{n_{\tau }}{\underset{j_{\tau }=1}{\sum }}\left[T_{c_{i_{\tau }}^{\tau }}^{t,j_{\tau }}P_{c_{i_{\tau }}^{\tau },j_{\tau }}^{t}\right]$$

### 2.3. Clumpr Package

We implemented an R package (https://github.com/UBESP-DCTV/clumpr, accessed on date 2 July 2021) along with a Shiny WEB interface (available at https://r-ubesp.dctv.unipd.it/shiny/clumpr/, accessed on date 2 July 2021) to provide a programmatic infrastructure and a platform to develop and monitor the probability model that reproduces the Italian system of the allocation of organs in “surplus”. The R Shiny app web interface allows the visualization of the results using customizable parameters, as shown in Table 2. Appendix A in supplementary report display data about the number of procured/transplanted/discarded lungs in 2015 and 2016.

The graphical WEB interface (Figure 2) is an easy tool that allows the input of the characteristics of interest according to the protocol, the national CRT network, and the role definitions.

The interface can currently show the probability that a center will accept a surplus organ, which is the ratio of the number of accepted organs and the overall number of organs offered to the center in the period considered. The other functionalities of the package are listed in the Appendix A.

## 3. Results

In order to evaluate the system of allocation of surplus organs, we used national data on lung transplantation from the years 2015 and 2016. The acceptance rate of each CRT was set at 26%, which is the mean of the acceptance rates between the years considered.

### The Equity of the Italian System of Surplus Allocation of Organ Transplants

Figure 3 and Figure 4 show the trends for the 2015 and 2016 and the probabilities according to the two models as the number of lungs exceeded increases. In both probability models, the Center-North MA seems to be favored. Lazio and Sicily, in the first probability model for both years, always possess a probability of less than 5% and yet another lung is offered to them. The northern regions have far higher probabilities for both years, especially in NITp which maintains a probability higher than 20%. After the second surplus lung, the probability for each region reached a plateau (Figure 2).

The second probability model (Figure 3) starts with a probability of 100% for each region with organ number one. After that, there is a progressive decline in the probability, which begins earlier in the southern regions up to the eigth surplus lung when the probability for the southern regions reaches 0 in 2015 and no further growth occurs. The same goes for the northern regions relative to the 18th organ and the 25th organ for NITp in 2015. In 2016, the mechanism was the same, except for the fact that the decrease started later for all the regions, especially the northern regions.

For exploratory purposes, we also set the acceptance rate at 90% and 100% since, theoretically, each center has no reason to reject an organ at first (Figure 5).

In the first probability model and up to the sixth organ (Figure 5), all regions have a zero probability at least once. After that, the probability for each region decreases with the growing lung numbers until everyone reaches a plateau after the 40th organ. By increasing the acceptance rate of the first probability model as the number of lungs offered increases, the probability reaches a plateau as shown by the horizontal lines, which are the median value of each probability and are comparable to the plateau reached in the same model in Figure 3. The second probability model (Figure 5, panels B and D) starts with a probability of 90% for each region with organ number one. After that, there is a progressive decline in probability, which begins earlier in the southern regions up to the 15th surplus lung when the probability for the southern regions reaches 0 and no further growth occurs. The same goes for the northern regions at the 23rd organ. The probability that the n-th lung is offered at a specific CRT decreases to zero when the maximum number of lungs produced for that year is reached an acceptance rate of 100%.

## 4. Discussion

In a framework in which there is a paucity of organ donors, it is of primary importance to focus on how to optimize their allocation. However, the equal and efficient allocation of this scarce resource is complex. Different healthcare systems have provided different answers to this problem, resulting in different results in terms of criteria for both the recipients and the donors [21]. In the European context, various countries have chosen to allocate organs first to the sickest patients following a different criteria [22], as in the Italian system. However, in an allocation procedure, the organ does not always fulfill the request of the recipient; therefore, it can be rejected. There are various reasons for the rejection of an organ, such as its clinical characteristics (organ quality and maintenance characteristics) and organizational characteristics. At this point, it is of primary importance to avoid the loss of the organ by increasing the communication among different centers since the time for reallocating the organ is limited. For example, the time for a CRT to evaluate a surplus organ is 45 min. For this reason, the allocation system must be efficient and so the relevance of monitoring its functioning when encountering inefficiencies is necessary to identify the limits and consequently make adjustments to the system.

The system of allocation of surplus organs reveals disadvantages for the southern regions in both probability models.

The discrepancies can be explained by the allocation mechanism present in the protocol and by the peculiarity of each CRT. The model is based on the concept of “local primacy” and so, at first, the organ is offered to patients in the local area of the organ donated [23]. The choice of offering the organ first to the MA of the donor is justified by the reduced distance that minimizes the cold organ ischemic time. The use of this concept benefits CRTs that produce more organs; as a matter of fact, in Figure 3 and Figure 4, the probability of receiving n-th lungs decreases later for CRTs that usually produce a greater number of organs. These CRTs usually belong to the MA-North. The Italian report on the donations and transplantation of organs in 2017 shows that donors are mainly from the northern regions, regardless of the type of organ considered (121 Piemonte, 142 Emilia Romagna, 167 Toscana, 130 Veneto, and 226 Lombardia against 50 Sicilia and 117 Lazio) [24]. These numbers may explain why northern CRTs, at specific points, maintain a high level of probability in the second model. Northern CRTs had a higher number of lungs transplanted [25]. Moreover, the satisfaction index of the waiting list (ratio between the number of transplants performed in a year and the number of patients on the waiting list in the same period) shows that the northern region is more efficient in the management of the waiting list [26]. This means that the northern regions not only produce a greater number of organs but also carry out a greater number of lung transplants. This last aspect influences the acceptance rate in the event that an organ is offered to an MA.

In order to better understand the system of allocations of lungs in Italy, it would be useful to define the probabilistic model underpinning all the protocols of allocation used, for example, the emergency and restitution protocols. More generally, Clumpr may allow the comparison of the Italian model with alternative organ allocation systems, such as those that use the LAS score [6]. The use of this score is already known in the Italian context, as it is already used for international organ exchanges in the European framework albeit in a modified version (“Italian Gate to Europe” program) [27] and also in the protocol of the NIT program [28].

Clumpr is suitable for modifications due to the flexibility and adaptability of its structure.

### Limitations

Clumpr uses only one of the Italian organ allocation protocols. The surplus protocol, however, only comes into play after the application of other protocols (urgency, list, return, and regional). Therefore, in order to comprehensively evaluate the equity of access to donated organs, it is necessary to implement the app while including other protocols. The app also does not show the probabilities for all the organs but only for the lungs. Finally, at the current stage of development, our results consider the same levels of acceptance. Future versions may remove this constraint by using the actual acceptance rate of each CRT.

## 5. Conclusions

The tool we developed is intended as a practical instrument to monitor the impact of the current allocation system of surplus organs.

The app permits the visualization of the model operating the protocol of allocation for surplus lungs. The evaluation of the probability of the k-th lung being allocated to one CRT rather than another CRT is useful for evaluating the stability of the system and for verifying whether there are any local disparities. The model that assesses the probability of receiving at least one organ is useful to evaluate the probabilities when increasing the number of surplus lungs and the model introduces some disparities. The Web is useful for decision makers because it allows them to visualize different scenarios when they want to make changes to the protocol (both at the functional level, e.g., rates of acceptance, or at the structural level, e.g., number of (macro-)regions, (macro-)areas, or strip positions).

Our results show that the impact of the allocation of the surplus organs depends mostly on the region where the organ is donated. This is in line with what has been found in the research literature. Italy has a national opposed donor rate that exceeds 30% and there is a discrepancy between the northern and southern regions in terms of the number of organs donated [24,29]. In the future, the re-defining of the protocol could be considered with other concepts of fairness and equity, such as the centers’ transplant volume or waitlist length. However, those criteria have to be harmonized with the concept of “local primacy” that underlies this protocol.

## Figures and Tables

**Figure 1 ijerph-18-07132-f001:**
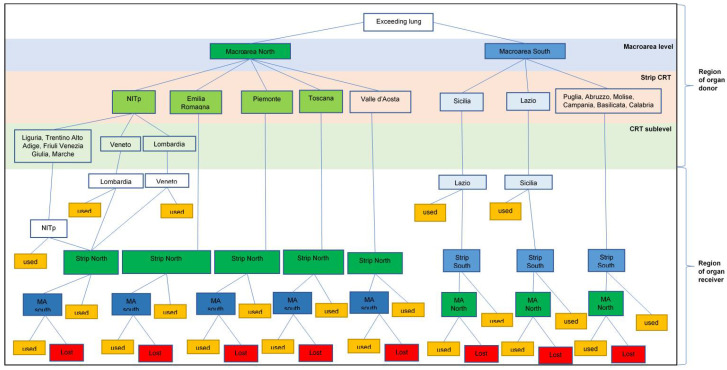
Flowchart of the allocation of exceeding lungs according to the national surplus protocol. The upper part refers to what happens in the donor region and the lower part shows what happens in the recipient’s location. In the upper part, there are two macroareas (north and south), which include a CRT on a regional basis and a grouping of several NITp regions. The CRTs highlighted in green and blue are those in which CRTs are present for the lung transplants and the regions without CRTs for lung transplants are grouped in boxes without any color.

**Figure 2 ijerph-18-07132-f002:**
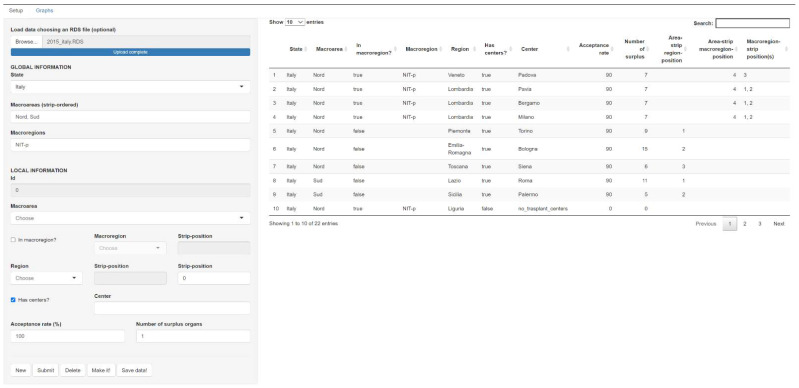
Web interface of the Clumpr app powered by the Shiny R package. The fields to be filled in are about the characteristics of interest defined in the protocol for the allocation of the surplus organs. The example shown in the screenshot reports the data inserted for the center in Padova with an acceptance rate of 90% and with 7 organs that were considered exceeding. The right side shows the other examples used for the probability graphs.

**Figure 3 ijerph-18-07132-f003:**
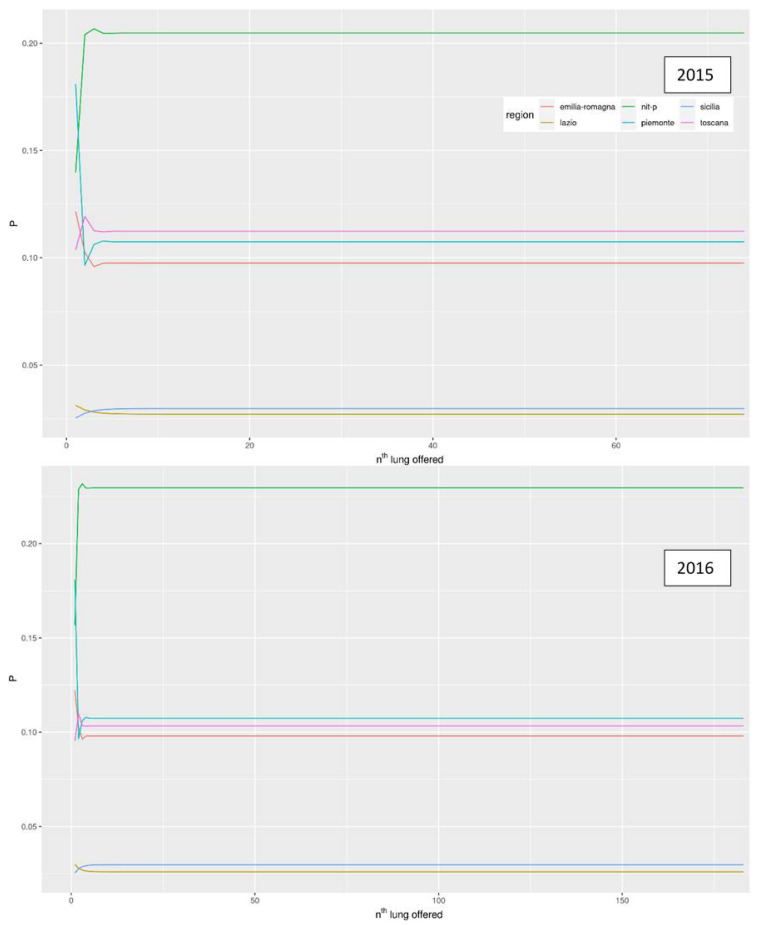
Trend of the probability for the n-th lung to be offered and accepted by each region as the number of lungs donated increases for the years 2015 and 2016.

**Figure 4 ijerph-18-07132-f004:**
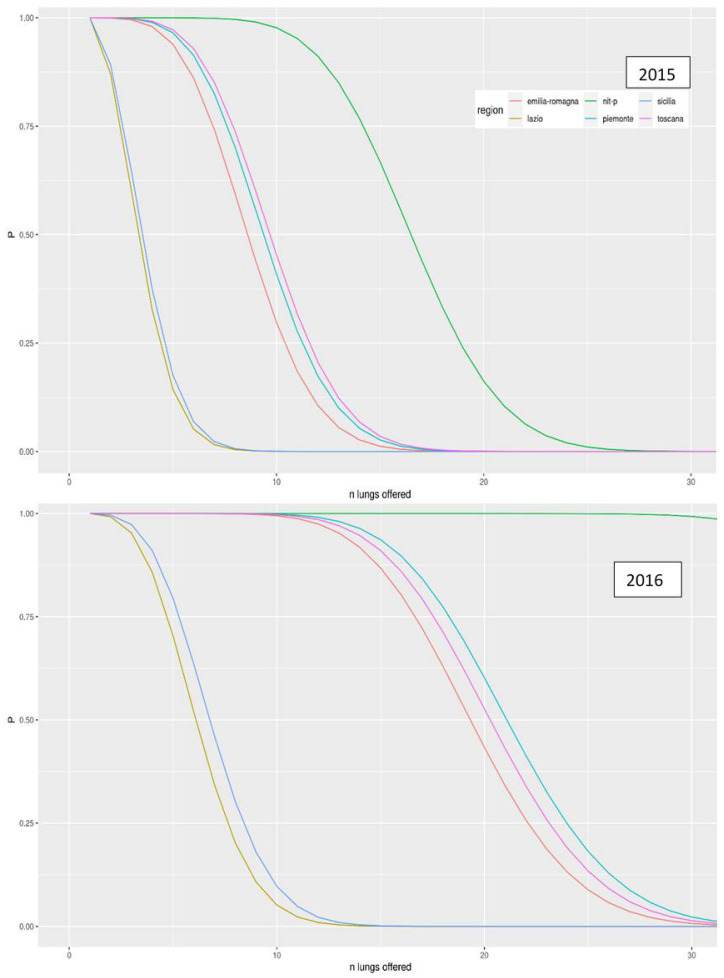
Trend of the probability that at least n-th lung will be offered and accepted by each region as the number of lungs donated increases for the years 2015 and 2016.

**Figure 5 ijerph-18-07132-f005:**
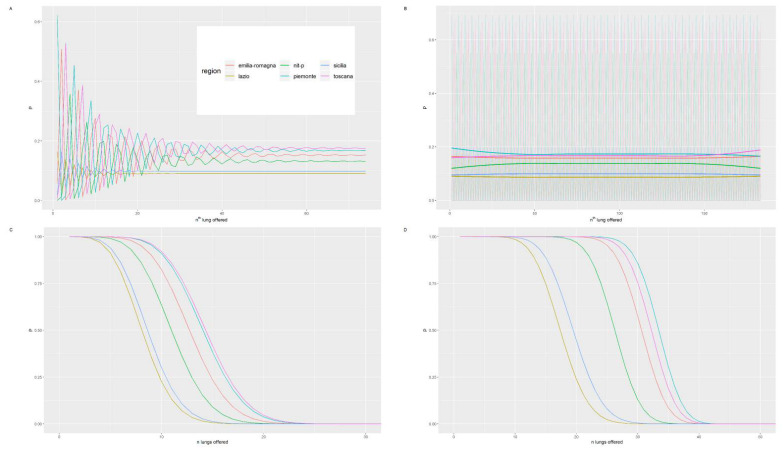
Panels (**A**,**B**) show trends of the probability for the n-lung to be offered and accepted by each region as the number of lungs donated increases with acceptance rates of 90 and 100%, respectively, in 2015. Panels (**C**,**D**) are the trend of the probability that at least n-th lung will be offered and accepted by each region as the number of lungs donated increases with acceptance rates of 90 and 100%, respectively, in 2015. In the background of panel B at the top, the trend of the probabilities can be observed, while the thicker lines depict the average of the probabilities for each CRT.

**Table 1 ijerph-18-07132-t001:** Strip continuous model as described in the “national protocol for the management of surplus organs in all transplant programs” for exceeding management of the lungs. Rotation of the strip will scale both the region that has accepted the surplus organ and all those regions that have rejected it. The order in the strip does not change if no one CRT accepts the offer.

**Macroarea North**			
Piemonte	Emilia Romagna	Toscana	NITp
**Macroarea south**			
Lazio		Sicilia	
NITp: North Italian Transplant program			

**Table 2 ijerph-18-07132-t002:** List of parameters required by the app for the calculation of both probability models: (1) The probability for the n-lung to be offered and accepted and (2) the probability that at least an n-th lung will be offered and accepted.

Parameter	Description	Type	Example
Macroarea	MA of organ origin	categorical	North
			South
Macroregion	Indication of macroregion when present	categorical	NITp
Region	Italian region with and without a CRT	categorical	Toscana
			Veneto
			Lombardia
Center	Indication of cities with a CRT	text	Siena
			Padova
			Bergamo
			Milano
Acceptance rate	Percentages of acceptation	number	26%
Number of surplus organs		number	3
NITp: North Italian Transplant program			

## Data Availability

Not applicable.

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
