# Peer review of "The Surplus Transplant Lung Allocation System in Italy: An Evaluation of the Allocation Process via Stochastic Modeling"

_ijerph, 2021, doi:10.3390/ijerph18137132_

Round 1

Reviewer 1 Report

The paper Corrado Lanera et al., titled "The Transplant lung allocation system in Italy: an Evaluation of the allocation process via stochastic modeling" concerns the procedure aimed to optimalise the allocation lung transplant in North and South Italy.

I think some question should be explained:

Please provide more information what the criteria were included for the optimising the allocation model in Italy.

What conditions/problem in Italy require the adaptation of the European system of allocation?

Figure 2 is unreadable.

Author Response

Open Review

English language and style

( ) Extensive editing of English language and style required
( ) Moderate English changes required
( ) English language and style are fine/minor spell check required
(x) I don't feel qualified to judge about the English language and style

Yes

Can be improved

Must be improved

Not applicable

Does the introduction provide sufficient background and include all relevant references?

( )

(x)

( )

( )

Is the research design appropriate?

( )

(x)

( )

( )

Are the methods adequately described?

( )

(x)

( )

( )

Are the results clearly presented?

( )

(x)

( )

( )

Are the conclusions supported by the results?

(x)

( )

( )

( )

Comments and Suggestions for Authors

The paper Corrado Lanera et al., titled "The Transplant lung allocation system in Italy: an Evaluation of the allocation process via stochastic modeling" concerns the procedure aimed to optimalise the allocation lung transplant in North and South Italy.

I think some question should be explained:

Please provide more information what the criteria were included for the optimising the allocation model in Italy.

We thank the reviewer for the careful consideration and overall positive judgement given to our work.

The work presented in this article is a description of the allocation system already existing. Therefore, the models presented at the moment are a mathematical translation of what already exists.

What conditions/problem in Italy require the adaptation of the European system of allocation?

Thank you for pointing this out.

In Europe, in the Eurotranplant network is used a modified version of the lung allocation system (LAS) to maximize the clinical matches between donor and recipient. In Italy there is no a unique method to define clinical matches. Only three lung transplantation centers in Lombardia region has introduced the LAS in 2016. The other centers are based on a “center decision”. The use of different score in Italy and Europe obviously require an adaptation in the defining how to match lung that comes from Italy to another European country (both int the Eurotransplant network and not) and viceversa. The use of the score it is possible will be introduced for exceeding lungs. Moreover, in a nation-wide urgent lung transplant programme has been established in Italy since 2010. This system is not existing in the country in the Eurotranplant network.

Added some information in lines 55-70

Figure 2 is unreadable.

Replaced as suggested.

Submission Date

06 May 2021

Date of this review

26 May 2021 09:07:37

Fine modulo

© 1996-2021 MDPI (Basel, Switzerland) unless otherwise stated

Reviewer 2 Report

  1. The phrase "exceeding organs" or "exceeding lungs" has been used a number of time in the manuscript. Probably, the intended phrase is "excess organs" or "surplus lungs".
  2. The paper would be of much more valuable if it could suggest a more improved method of distributing donated organs for transplant to achieve optimized results. 

Author Response

Reviewer 2

Open Review

English language and style

( ) Extensive editing of English language and style required
( ) Moderate English changes required
(x) English language and style are fine/minor spell check required
( ) I don't feel qualified to judge about the English language and style

Yes

Can be improved

Must be improved

Not applicable

Does the introduction provide sufficient background and include all relevant references?

(x)

( )

( )

( )

Is the research design appropriate?

(x)

( )

( )

( )

Are the methods adequately described?

(x)

( )

( )

( )

Are the results clearly presented?

( )

( )

(x)

( )

Are the conclusions supported by the results?

( )

( )

( )

( )

Comments and Suggestions for Authors

We thank the reviewer for the careful consideration and overall positive judgement given to our work.

  1. The phrase "exceeding organs" or "exceeding lungs" has been used a number of time in the manuscript. Probably, the intended phrase is "excess organs" or "surplus lungs".

Thanks for the suggestion.

Modified as suggested through the manuscript.

  1. The paper would be of much more valuable if it could suggest a more improved method of distributing donated organs for transplant to achieve optimized results. 

Thanks for pointing this out. The work presented in this article is a description of the allocation system already existing. Therefore, the models presented at the moment are a mathematical translation of what already exists. On the other hand, the Web application allows to simulate different scenario (both at the functional level, e.g., rates of acceptance, or at the structural level, e.g., number of (macro-)regions, (macro-)areas, or strip positions). When applying changes at the protocol it will be possible to simulate in the app before the direct application of the changes.

Added lines (549-551).

Submission Date

06 May 2021

Date of this review

06 Jun 2021 17:46:33

Author Response

Reviewer 3

Open Review

English language and style

( ) Extensive editing of English language and style required
(x) Moderate English changes required
( ) English language and style are fine/minor spell check required
( ) I don't feel qualified to judge about the English language and style

Yes

Can be improved

Must be improved

Not applicable

Does the introduction provide sufficient background and include all relevant references?

( )

( )

(x)

( )

Is the research design appropriate?

( )

(x)

( )

( )

Are the methods adequately described?

( )

( )

(x)

( )

Are the results clearly presented?

( )

( )

(x)

( )

Are the conclusions supported by the results?

( )

( )

( )

( )

Comments and Suggestions for Authors

Submission Date

06 May 2021

Date of this review

13 Jun 2021 00:41:49

We thank the reviewer for the careful consideration and overall positive judgement given to our work.

  1. The paper aims to develop stochastic models to evaluate and describe the allocation of

surplus lungs in Italy. However, the main manuscript does not propose any analytical

models in detail. Models, propositions, lemma, and corollaries have been placed in the

supplementary materials, whereas they should have been embedded in the main

manuscript. I suggest authors create a section titled "Model" and place all the relevant

analytical models, propositions, etc. in this section. The proof of main propositions, corollary,

etc. should be included in the appendix.

Thanks for the suggestion. Added the relevant analytical models, propositions, etc in the section 2.2. Probability models (pag 4-6)

  1. The literature review needs major improvement. Despite existing rich literature in organ

allocations, there is almost no proper literature review in this paper. The authors should

devote a section to appropriately review both analytical and non-analytical types of research

done worldwide in organ allocation in general and more specifically in lung allocation.

Thanks for the suggestion, updated the literature.

  1. The authors need to clearly state the existing research gap and the need for their research

to fill the gap? What is the practical implication of this research? In other words, how do

this research and the developed app benefit the waitlisted patients, organ transplant centers,

and organ allocation agencies?

The aim of this research is to formally present the Italian surplus transplant protocol, providing a description, and a Web application. The web application permit decision makers to explore the current or new scenarios and thus to modify or develop a new protocol to enhance the programme itself. The ability to explore the current scenario is helpful to evaluate if there is inequality in terms of organs allocation.

4.The introduction section needs to be revisited. The authors should clearly state the main

contribution, the novelty, and the importance of their research.

The main contribution of this work is to “provide an interactive visualization tool for evaluating whether the current Italian policy on organ distribution can potentially lead to distortions or inequalities in organ allocation systems across different areas of the country.” (lines 70-75).

  1. The references are very outdated and not sufficient. The authors need to include the most

recent papers published in recent years (2019, 2020, 2021). For years before 2019, more

appropriate references should be provided. The paper has only 19 citations despite the

excessive amount of literature that exists in the area of organ transplantation worldwide.

Thanks for pointing this out. Updated the literature.

  1. The national data on lung transplantation used in this research to assess the allocation of

excess donor lungs are not quite new (years 2015 and 2016). Do the authors have access to

the newer data? Also, the authors do not provide enough information and statistics on the obtained data. What type of information is included in the dataset, e.g., donors' clinical characteristics (history of high blood process, diabetes, heart disease, lung disease,..), demographic (race, gender,..), behavioral (smoking,..) and geographical (regions, city, zipcode,..)?

How many records and features are included in the data set? How many lungs

got procured/transplanted/discarded and were available during the study period?

In this work we are presenting how the Italian surplus transplant protocol work. We are describing the function of the system in organizational and logistical aspect, so we are not considering the characteristics of the donor. In a different work it will be of interest to evaluate, as suggested by the reviewer, the characteristics of the donor when considered as a surplus. So, in this work there is not a datasets with “records” of patients characteristics. Data about the number of procured/transplanted/discarded lungs were now available in the supplementary material.

  1. The authors need to provide useful statistics on both waitlisted and transplanted patients

across all regions. For instance, how many patients are waiting to receive lungs across all

regions? How many patients die or get removed from the waitlist due to health

deteriorations across all regions? How many transplants happen across all regions? Etc.

Thanks for the suggestion. Data about the number of procured/transplanted/discarded lungs were now available in the supplementary material. Other data are not of interest for this work. Moreover, as suggested the inclusion of these kind of data would be useful to implement this work focusing on clinical aspects rather than organizational.

  1. The authors need to provide more detailed information on donor clinical characteristics

when it comes to the surplus lungs. For instance, are surplus lungs are coming from old

donors or donors with underlying health issues? What percentage of surplus lungs are

marginal lungs(lungs with low quality)?

This work is a description of how the Italian surplus transplant protocol work. So, there is no need of donor characteristics for the purpose of this work. Nor, any data at donor-level was used.

  1. How does the lung allocation process (not the surplus lung allocation) work in Italy? The

information provided in lines 55 to 60 and 69 to 70 does not suffice, and I could not figure

out the details of deceased lung allocation in Italy. Are there multiple waitlists? For instance,

do multiple regional waitlists and one national waitlist exist? So if an organ retrieved in any

specific region is failed to find a recipient or proper match, would it be offered to the next

closest region or nationwide or considered a surplus organ?             

Thanks for pointing this out. Added description of the Italian system in the introduction as suggested (lines 65-80).

  1. As the paper explains, the donor's lungs are considered "surplus" when "the organ is not

allocated through the national program of urgency and is not used in the location of the

donor ". Does Cold Ischem Time (CIT) affect the definition of the surplus organ? In other

words, do organs get discarded before being identified as surplus organs due to long CIT

and its detrimental effect on the lungs' qualities?

No, Cold Ischem Tune does not affect the definition of  a “surplus” .

  1. How do fairness and equity are considered when it comes to distributing the surplus

organ? For instance, besides geographical consideration explained in Figure 1, should be

any correlation between centers' transplant volume or waitlist length and the probability of

receiving a surplus organ?

Thanks for pointing this out. The surplus organs follow the geographical scheme explained in Figure 1. Waitlist length and centers’ transplant volume were not considered when allocating a surplus organ. Those criteria may be element to be take on count when re-defining the allocation of a surplus organ However they be harmonised with other concepts that underlies this model, such as the concept of “local primacy”. “The use of this concept benefits CRTs that produce more organs; as a matter of fact, in Figure 3 and Figure 4, the probability of receiving n-lungs decreases later for CRTs that usually produce a greater number of organs.”

  Added some lines (436-439).

Can you provide any statistics to compare the number of surplus organs and their qualities (e.g, old donors, donors with specific health issues concerning transplant outcomes) across organ donation centers and transplant centers in Italy?

As reported before, this work describe how the Italian surplus transplant protocol works. So there is no need of statistics related on the quality of the surplus organ.

Suppose the surplus organs are mostly of lower quality which is why no one accepted them in the

donor region, would it be fair to offer those organs to regions/ centers that have a strong

history of turning down marginal organs?

The surplus organs could be considere a “surplus” both because does not fulfil the request of the recipient or for organizational characteristics. So, there is a probability, by definition, that the surplus organ could be of lower quality. In this work we have not considered why an organ is considered surplus since it does not influence the allocation system.

  1. Some figures presented in this paper do not have good quality and are not clear. They

need to be replaced.

Thanks for the suggestion. Replaced the figures that have a low quality.

  1. The notations can be very confusing and hard to follow to the readers. The authors used

excessive subscripts and superscripts (e.g., propositions 1.1, equation 2, 3a, 4,9, ...).

In the section Definition 1.1 in the supplementary material (S3) describe all the notations, which are consistently maintained across all the mathematical formulas used in the manuscript. It has been developed to be unambiguous, clear and minimal. We agree with the reviewer that it can be a little hard to follow, but we selected it precisely against the option of adopting a notation which would be more verbose and ambiguous at the same time.

Round 2

Reviewer 2 Report

The authors have completed the suggested changes satisfactorily.

Reviewer 3 Report

Most of my concerns from my previous review have been addressed.